# Heterogeneity of Colorectal Cancer Progression: Molecular Gas and Brakes

**DOI:** 10.3390/ijms22105246

**Published:** 2021-05-15

**Authors:** Federica Gaiani, Federica Marchesi, Francesca Negri, Luana Greco, Alberto Malesci, Gian Luigi de’Angelis, Luigi Laghi

**Affiliations:** 1Department of Medicine and Surgery, University of Parma, 43126 Parma, Italy; federica.gaiani@unipr.it (F.G.); gianluigi.deangelis@unipr.it (G.L.d.); 2Gastroenterology and Endoscopy Unit, University-Hospital of Parma, via Gramsci 14, 43126 Parma, Italy; 3IRCCS Humanitas Research Hospital, via Manzoni 56, 20089 Rozzano, Italy; federica.marchesi@humanitasresearch.it (F.M.); alberto.malesci@hunimed.eu (A.M.); 4Department of Medical Biotechnology and Translational Medicine, University of Milan, 20132 Milan, Italy; 5Medical Oncology Unit, University Hospital of Parma, 43126 Parma, Italy; fnegri@ao.pr.it; 6Laboratory of Molecular Gastroenterology, IRCCS Humanitas Research Hospital, via Manzoni 56, 20089 Rozzano, Italy; luana.greco@humanitasresearch.it; 7Department of Biomedical Sciences, Humanitas University, Via Rita Levi Montalcini 4, 20072 Pieve Emanuele, Italy

**Keywords:** colorectal cancer, progression, heterogeneity

## Abstract

The review begins with molecular genetics, which hit the field unveiling the involvement of oncogenes and tumor suppressor genes in the pathogenesis of colorectal cancer (CRC) and uncovering genetic predispositions. Then the notion of molecular phenotypes with different clinical behaviors was introduced and translated in the clinical arena, paving the way to next-generation sequencing that captured previously unrecognized heterogeneity. Among other molecular regulators of CRC progression, the extent of host immune response within the tumor micro-environment has a critical position. Translational sciences deeply investigated the field, accelerating the pace toward clinical transition, due to its strong association with outcomes. While the perturbation of gut homeostasis occurring in inflammatory bowel diseases can fuel carcinogenesis, micronutrients like vitamin D and calcium can act as brakes, and we discuss underlying molecular mechanisms. Among the components of gut microbiota, Fusobacterium nucleatum is over-represented in CRC, and may worsen patient outcome. However, any translational knowledge tracing the multifaceted evolution of CRC should be interpreted according to the prognostic and predictive frame of the TNM-staging system in a perspective of clinical actionability. Eventually, we examine challenges and promises of pharmacological interventions aimed to restrain disease progression at different disease stages.

## 1. Heterogeneous Gene Damage and Different Progression of Colorectal Cancer

### The Discovery of the Main Patterns of Gene Damage, CIN and MSI

More than three decades ago, molecular genetics introduced a substantial revolution in oncology. The discovery of the derangements due to the activation of oncogenes and the silencing of tumor suppressor genes put forward the notion that the type and amount of genetic damage underlie the development and the evolution of cancer. In the beginning, hunting for genes responsible for inherited predispositions to cancer had a pivotal role, and colorectal cancer (CRC) acted as a key model. The discovery that *APC* was the gene underlying the development of familial polyposis (FAP), once mutated in the germline [1,2,3], fit the two-hit hypothesis by Knudson [4] in light of somatic inactivation of the other allele in somatic tumor cells [5], mostly due to loss of heterozygosity, as in the case of retinoblastoma [4]. It was also shown that the same type of gene damage occurs in most sporadic tumors, inactivating the two alleles of this gatekeeper in somatic cells along colorectal carcinogenesis. In the early 1990s, the main players in CRC were *APC*, *KRAS* [6] and *TP53* [7,8], and a model for their mostly sequential multistep damage [9] became a paradigm in cancer genetics [10]. Meanwhile, moving from the APC-FAP lesson, researchers were trying to identify the culprit for hereditary non-polyposis colorectal cancer (HNPCC; now referred to as Lynch syndrome) [11,12]. Independent teams contributed to the discovery of the molecular phenotype of this disease, known as microsatellite instability (MSI), although the initial terminology differed according to the groups engaged in the competitive discovery [13,14,15]. It became rapidly appreciated that Lynch syndrome and MSI arose because of germline defects in one of the genes of the DNA mismatch repair (MMR) system (namely *MLH1*, *MSH2*, *MSH6*, *PMS2* and *EPCAM* deletion) [16]. Additionally, MSI observed in a relevant fraction of sporadic CRCs (≥10%) was short after being linked to somatic silencing of *MLH1*, due to its promoter hypermethylation [17,18,19]. Altogether, MSI inherited and sporadic cancers account for 15–20% of CRCs, and their somatic damage is different from that of non-MSI CRCs (or MS-stable, MSS). MSI cancers harbor thousands of unrepaired replication errors, mostly frameshift mutations not observed in MSS tumors, which otherwise display much higher degrees of chromosomal damage (from rearrangements to aneuploidy), and were thus also termed chromosomally unstable or CIN (Table 1). The growing importance of understanding that a molecular classification of CRC could be attained is exemplified by the appearance of the feasibility of a molecular screening that could allow distinguishing MSI from MSS CRCs at end of the 1990s [20]. Surprisingly, clinical actionability of such molecular differentiation, although well established, was unanimously recognized by scientific societies active in the clinical arena 15 years later [21]. Expanding the lessons learned from inherited predispositions, the molecular heterogeneity of CRC was becoming apparent, spreading the notion that this disease encompasses entities with different progression (i.e., natural history) and postsurgical evolution [22]. Clinically, it became increasingly appreciated that MSI cancers display a significant better postsurgical outcome, largely explained by the low rate of patients presenting with advanced disease at diagnosis, due to their reduced metastatic potential [23,24,25]. MSI and CIN cancers were also differentiated according to the responsiveness to cytotoxic chemotherapy [26], thus anticipating the notion that the type of genetic damage may modify the responsiveness to drugs as well [27], and later it was shown that defective MMR is a predictive marker for lack of efficacy of fluorouracil-based adjuvant therapy in CRC [28].

More recently, the responsiveness of progressive metastatic CRC to immune checkpoint blockade (anti-programmed death 1 immune check point inhibitor) was shown to occur more frequently in patients with MMR-deficient cancers (harboring an average of 1782 somatic mutations; see below) than in MMR-proficient cancers (harboring a mean of 73 mutations) [29].

Thus, molecular data over time have established the bases for the notion that different types of genetic damage underlie different natural histories of CRC progression, as well as its postsurgical outcome and drug-responsiveness.

## 2. DNA Hypermethylation and CIMP

As the clinical–pathological differences between MSI and MSS CRCs were becoming increasingly evident, a further molecular subtype was being investigated. Moving from the methylation status of *MLH1* in sporadic MSI CRCs [30], it was appreciated that gene hypermethylation events cluster in a fraction of cases, overlapping with sporadic MSI [31] ones due to *MLH1* hypermethylation. The comparison of the increased frequency of epigenetic events at certain loci (although these were not as well standardized as those tested to establish MS-status) coincided with the proposal of a third CRC molecular subtype, referred to as CpG island methylator phenotype, or CIMP [32,33] (Table 1). Promoter hypermethylation leading to gene silencing would thus resemble other gene-silencing mechanisms, and also can occur as a second hit in genes like *APC* [34]. Similar to MS typing, the CIMP profile obtained by the analysis of given loci allows differentiating CRC accordingly (i.e., CIMP high vs. low vs. no-CIMP). CIMP+ or high CRC had a peculiar profile [35], associated with older age, proximal location, poor differentiation, MSI-high and *BRAF* mutation [33], and inversely with *LINE-1* hypomethylation. CIMP-high CRCs were also found to have a better outcome than CIMP-low, particularly if showing wild-type *BRAF* [36]. The concept that was originating was that it would be eventually possible to reach a molecular pathological epidemiology of CRC exploiting molecular classification and incorporating interactions with environmental factors, as well as associations with clinical outcome [37].

## 3. The Advent of Next-Generation Sequencing and the Evidence of Widespread Genetic Heterogeneity

While these classification schemes [36] were being variably adopted in translational research [38], new sequencing technologies (i.e., next-generation sequencing, NGS) hit the research ground, allowing an unbiased identification of the extent of genetic damage in cancer [39], which was previously unthought. These innovative explorations showed that the average number of gene mutations in CRC was approximately 80, of which one out of 5–6 would occur in candidate cancer genes. It also emerged that such candidates encompassed genes for which, in spite of functional studies, no mutational evidence had been previously reported for their association with cancer, as well as genes not previously linked to neoplasia. Such candidates comprised transcriptional regulators, genes involved in cell adhesion and signal transduction. The heterogeneity of mutated genes was exemplified by the shared number of candidate cancer gene mutations, not exceeding six common mutants among cancers. These notions were refined shortly afterward by drawing the genomic landscape of CRC [40], which, when recapitulating these results, showed how a few mutational peaks (or “mountains”) in known cancer genes are outnumbered by a multitude of hills represented by infrequently mutated genes. The previous focus on mountains was largely determined by available technology, while NGS introduced new paradigms. In this novel mutational milieu, a minority of the events is responsible for driving the processes of tumor initiation, progression and maintenance. The vast heterogeneity of the mutational hills occurring in individual CRC could still be recapitulated by the pathways they derange. Thus, it could be possible to classify the main alterations occurring during tumorigenesis according to the pathways targeted by mutational events. Along this line, mRNA sequencing by NGS provides a way to identify the alterations of gene expression occurring in colorectal carcinogenesis, and by mean of this approach, an international consensus was thus proposed comprising four molecular subtypes (i.e., CMS1 to CMS4) [41]. This network-based approach used aggregated expression data from six previously analyzed cohorts [41], and eventually recapitulated CRC subtypes into MSI immune (CMS1), canonical (CMS2), metabolic (CMS3) and mesenchymal (CMS4) (Table 1). This taxonomy was based upon differences in gene expression, mainly refining the classification of non-MSI subtypes. These expression patterns also reflected in individual clinical behaviors marked by different relapse-free survivals and survival after relapse. However, gene-expression patterns are influenced by their stromal content, which contributes to the type and quantity of detected transcripts. Isella et al. showed that this is the case for the mesenchymal subtype, and that transcriptional signatures incorporating cancer-associated fibroblasts (CAF), leukocytes or endothelial cells were more abundant in CRC classified as mesenchymal [42]. Interestingly, CRC with a high content of CAF transcripts was associated with a worse outcome, specifically in the absence of adjuvant therapy. Accordingly, an evolution of the classification employing transcriptional signatures was then developed following the depletion of the stromal signatures, which can be obtained by xeno-transplantation. This approach assessing intrinsic translational features of cancer cells led to the identification of five CRC intrinsic subtypes (CRIS; A to E), in which transcriptional signatures are inherent to neoplastic cells deprived of the stromal components [43] (Table 1). As this classification was experimentally developed by moving from CRC samples that had produced liver metastases, it might better fit aggressive tumors than those with smolder behavior. These studies testify that together with technological improvement, bioinformatics entered into the arena of molecular analysis, modifying the classic “black and white” or null hypothesis approach. Clearly, overlaps exist among the different classification schemes, and certain historically proven paradigms persist, chiefly the taxonomic independence of MSI/CIMP/BRAF-mutated tumors. Differently, the stromal contamination may affect the independence of a mesenchymal subtype, thus questioning the occurrence of epithelial to mesenchymal transition (EMT) in CRC [44]. At any event, taxonomic features like the content of CAF signatures remain a negative prognostic factor, indicating the relevant contribution exerted by the stromal compartment in determining disease progression.

Under several respects, it became progressively evident that intrinsic genetic and epigenetic features of the tumor are not the only factor that can explain the different behaviors of CRC. While the type of gene damage inherently drives the evolutive speed of cancer, other “extrinsic” processes are involved in determining its progression. Among these is the immune response of the host, comprising chiefly its adaptive immune arm [45], but not restricted to it [46,47]. The playgrounds for cancer restraint or fueling could be local; i.e., the tumor microenvironment (TME), as well as systemic and at distant sites, such as the metastatic niche [48].

## 4. Tumor-Host Immune Response as Switcher on the Routes of Cancer Progression

Alongside more common histopathological and molecular classifiers, recent years have witnessed the emergence of immune components as prognostic markers in CRC [45,49,50]. What is commonly referred to as the immune contexture [51]; i.e., the density and types of immune cells infiltrating cancer tissues, has been object of studies aimed at both high-resolution definition (primarily achieved with multidimensional approaches) and narrowing down to specific biomarkers to be used in daily routines. The Immunoscore represents the ultimate output of those studies [52,53].

Efforts aimed at providing associative links between specific immune cell types and distinct disease outcomes set their foundations on earlier observations that most cancer tissues host immune cells in their microenvironment [54,55], and on mechanistic evidence of the involvement of immune-based circuits in cancer progression [56,57,58,59,60]. Particularly relevant have been studies aimed at showing the causative link between inflammation and cancer occurrence and progression [56,60]. On the other hand, the contribution of adaptive immunity to recognition and elimination of cancer cells has been known for a long time [54,55]. Both components, innate and adaptive, with their complex and intersecting protumor and antitumor capabilities clearly emerge from deep analyses of the microenvironment of CRC [61]. A balance between the two is likely to contribute to progression versus resistance.

Human studies have not allowed, so far, to mechanistically define the sequence of events that cause accumulation of specific immune subsets in cancer tissues. Despite the fact that recent high-dimensional studies have shed light on the variety of immune cells in human CRC tissues [61], fully elucidating the complex dynamics and relative contribution of resident versus recruited immune components requires further studies. Nonetheless, a general scenario depicting how immune cells infiltrate a tumor is represented by the cancer immune cycle [62], according to which antigen-presenting cells, mostly dendritic cells (DCs), infiltrate the tumor tissue, uptake tumor-derived products by various innate recognition receptors, produce type I interferons and traffic to draining lymph nodes, where they present antigens to antigen-specific cytotoxic T cells (Figure 1). This event may be more efficient in tumors expressing neoantigens or with a high mutational burden [63]. Subsequent migration of activated T cells through the circulation and back to the tumor, guided by chemokine gradients including primarily CXCL9/CXCL10, would account for the high density of T cells in cancer tissues [64]. As to the activation status of T cells, prolonged immunosuppressive circuits, such as inhibitory axes like CTLA-4 and PD-1, may be responsible for T-cell dysfunction, accounting for the immune escape and cancer progression.

As mentioned, resident immune populations; e.g., tissue resident macrophages (TRMs) or intraepithelial lymphocytes (IELs), variably contribute to the balance of protumor or antitumor functions. Macrophages are specialized phagocytes with a high capability to ingest cellular debris, present antigens and impact on the adaptive immune response through cytokine production [65]. Their plasticity is a peculiar feature, whereby they can adopt an inflammatory phenotype ensuing in tumor elimination, as well as mature to subtypes evidently engaged in protumor functions. In the colon, TRMs have been described as constantly replenished by circulating monocytes [66]. This peculiarity that distinguishes them from other, long-lived TRMs may account for the exceptional favorable prognosis associated with macrophages in human CRC.

## 5. Cellular and Molecular Players in the Tumor Microenvironment: Meaningful Links

The immune microenvironment of CRC has gained much attention in the last few years, primarily because of the coexistence of protumor inflammatory signals and antitumor adaptive immune responses. These two almost opposite scenarios impinge into distinct clinically relevant outcomes. The link between chronic inflammation and CRC is robustly reflected in a higher risk of malignant transformation in inflammatory bowel disease (IBD) patients [60,67,68,69,70]. On the other hand, the strong capability shown by T-cell-related variables to stratify CRC patients in prognostic groups [45,49] suggests the existence of effective antitumor adaptive circuits. Translation of this knowledge to evidence-based biomarker identification is an active field and holds promise for better management of CRC patients. Both soluble mediators and cell types are being evaluated as markers of disease progression, based on mechanistic evidence of their involvement in the TME of CRC.

### 5.1. Soluble Mediators

The considerable and persistent release of inflammatory mediators in the TME is causatively linked to the strong association between IBD and CRC development [68,71,72,73]. Persistent infections [60,74], as well as sterile tissue damage (leading to release of alarmins, cell-stress signals, free nucleic acids), are acknowledged as drivers of the inflammatory response, by generating molecular patterns recognized as harmful by innate inflammatory cells [58,59]. Activation of key transcription factors, such as NF-kB and STAT-3, critically induces production of inflammatory mediators, including interleukin 1 beta (IL-1β), tumor necrosis factor-alpha (TNFα), interleukin 6 (IL-6) and chemokines (CCL2 and CXCL8), further fueling recruitment of inflammatory leukocytes. Both cytokine mediators with a clear tumor-inhibitor effect, such as interferon-gamma (IFN-ɤ); IL-12, 15 and 18 [73]; and a protumor one, such as IL-6, IL-17A, IL-22 and IL-23, have been recorded in CRC [73,75]. For others, such as IL-1 and TNFα, which are master inflammatory cytokines, the role is still debated and highly dependent on the experimental setting [71]. Collectively, the divergent roles of cytokines in CRC could be explained by the coexistence of some inflammatory mediators orchestrating specific antitumor immunity [71,76] and a variety of cytokines sustaining and fueling detrimental protumorigenic inflammation. The critical contribution of these players and of other innate mediators, such as pentraxin-3 (PTX3) and C reactive protein (CRP), involved in early inflammatory circuits to the inflammatory milieu, have promoted studies aimed at testing their prognostic value in CRC [73,77,78,79]. Blood markers of oxidative stress have been found to be strongly associated with poor prognosis in CRC [80]. AN emerging concept is that profiling of multiple cytokines is a better approach, based on evidence that protumorigenic and antitumorigenic cytokines are found and correlate with disease outcome [76].

### 5.2. Immune Cell Players

The occurrence and clinical relevance of effector T cells in CRC has enjoyed a lot of attention in the last decade, due to the already-discussed translational implications [45,49,50,81]. In a recent study on CRC, Zhang et al. finely profiled immune subsets using a comprehensive sequencing approach and identified 20 clusters of T cells, of which eight were CD8^+^ and 12 were CD4^+^ T cells [82], allowing concomitant tracking of trajectories of some T-cell subsets to others. These approaches open a range of possibilities to gain more insights into immune infiltrating cells, with the potential to be translated to identification of relevant markers.

Regulatory T cells (Treg) are essential suppressive modulators of intestinal inflammation, thanks to their production of the anti-inflammatory cytokines IL-10 and transforming growth factor beta (TGF-β); therefore, they were supposed to be impairing anti-tumor immune responses. Instead, solid evidence of an association of Treg density with favorable prognosis [83,84,85] in human CRC suggest that they may in fact be beneficial for the restriction over protumorigenic inflammation [86].

Attention to B-cell infiltration in CRC has been raised by their frequent occurrence within tertiary lymphoid structures, organized environments of T and B cells commonly associated with favorable prognosis across cancers, including CRC [87,88,89]. Nonetheless, B cells infiltrating cancer tissues as scattered cells have been shown to possess protumor functions in other malignancies [90,91], suggesting that the local organization of B lymphocytes is an important feature, with impact on their function and prognostic significance.

Macrophages, the most abundant immune cells within the CRC microenvironment, have the capability to modulate every step leading to carcinogenesis and tumor progression [58,92,93,94]. In CRC, macrophages are orchestrators of an inflammatory milieu considered a driver of tumor initiation and progression [95]. Single-cell analyses identified six macrophage subsets, including two clusters of TAMs enriched in tumors and three clusters of recruited macrophages [61]. Despite this clear engagement in protumor functions, studies aimed at defining the prognostic role of macrophages in CRC have shown surprisingly association with favorable prognosis [96,97,98] and response to therapy [99].

## 6. Links between Genetic Changes and the Immune-Contexture

Clearly, a link exists between genetic damage and the host immune response, and again the lesson comes from MSI CRCs, which have long been known as deeply infiltrated by T cells [100]. Such association also led in the past to the inclusion of a dense immune infiltrate (as in Crohn’s colitis) among the criteria advocated for MSI testing [101], long before universal screening for MMR defects were endorsed. High immunogenicity of MSI CRCs is sustained by their defective MMR, which results in large amounts of truncated peptides [102,103,104], acting as neo-antigens [105]. Thus, dense tumor infiltrating lymphocytes (TILs) are a sort of twin of most MSI CRCs [106], and sustain the associative link with better outcomes for this tumor type [49]. Yet, it is worth noting that high TIL amounts may not be the only reason for such a prognostic link. In MSI CRCs, the lack of CIN and of relevant damage in TSG is coupled with the peculiar genetic damage ensuing from the mutator that mutates other mutator pathways [103], which may not undergo the same type of selective pressure that pushes toward the enrichment of aggressive clones in MSS CRCs. MSI tumors are not the only ones significantly associated with dense TILs and better outcomes, as MSS CRCs with pathogenic somatic mutations in the *POLE* proofreading domain also share both high TILs and good outcomes [107,108]. Accordingly, “ultramutated” CRCs have a different clinical behavior, dictated by the type of genetic damage (whether it originates in the germline or in somatic cells), and sustained by the amount of adaptive immune reaction that they elicit.

It would be advisable to link the classifications pursued by DNA and mRNA data with those obtained by typing infiltrating immune cells, which include TILs but also innate cells, chiefly macrophages [109]. Such classification effort is meaningful, looking at patient outcome in various settings that should move from stage at diagnosis and include treatment [110]. An interesting paper published by Giannakis and colleagues joined the assessment of TILs with NGS analysis [111]. They found that even within MSS CRC, a high TIL amount correlates with high loads of neo-antigens. Other associations were with mutations in HLA genes and in members of the antigen-processing machinery.

Immune cells in the microenvironment of human CRC significantly correlate with postoperative tumor progression and response to therapy, fostering the development of new immune prognostic tools and increasing our ability to stratify patients into clinical subgroups. Most of the work done until now has focused on histopathological assessment, while high-resolution technologies are rapidly unearthing the complexity and diversity of immune cells in cancer tissues. In nonmetastatic settings, one could look for TNM together with biomarkers that could allow the prognostication of CRC cured from surgery alone and the prediction of responsiveness to adjuvant treatment of those requiring postsurgical therapy [110].

## 7. Inflammation as an Accelerator of Carcinogenesis in Inflammatory Bowel Diseases

The crucial role of immune hyperactivity and inflammation as cancer promoters is also exemplified by the increased risk of cancer in IBD patients, which is historically acknowledged, both for those affected by ulcerative colitis (UC) and Crohn’s disease (CD). Therefore, the theme of cancer surveillance has become of growing importance in recent decades, in terms of early diagnosis, understanding of the mechanisms for carcinogenesis and awareness of the risk factors concerning this particular group of patients, including mucosal inflammation and long-term immunosuppression [58]. From an epidemiologic point of view, cancerous lesions usually develop in the adult age, but the adaptation of the treatment and the optimization of the management are of paramount importance for the long course of the disease, starting from the pediatric age.

With respect to the basic risk of developing specific cancers, pathogenesis and epidemiology, literature data are conflicted. Patients affected by UC included in a Finnish study demonstrated an increased risk of colon, rectal, biliary tract and thyroid cancers, with the risk of CRC being highest among the youngest patients. Patients with CD had a significantly increased risk for cancers of the small intestine, anus and biliary tract, and also for myeloma [112]. In contrast, data from Denmark indicated that only CD patients had an increased risk of developing malignancies overall, such as small bowel cancer, lung cancer or non-Hodgkin’s lymphoma, while the general risk for developing cancer in UC patients was not increased [113]. Again, a large population-based study using Danish healthcare databases found that patients with IBD, particularly CD, were at an increased risk for gastrointestinal and extraintestinal malignancies [114]. One pediatric French population-based study estimated the risk of cancer in patients with childhood-onset CD (median age at diagnosis 14.6 years; median follow-up 11.4 years), and found a significant 2.5-fold increase compared with the background population [115]. A similar two-fold significant increased risk of cancer was also described in a Danish study that evidenced an overall risk of cancer in the population diagnosed at the age of 19 years or less of 2.17-fold, compared with the non-IBD population, and was the highest among the other age groups [116]. Overall, IBD are well-recognized risk factors for the development of colorectal and small bowel cancer; in particular, UC and colic CD are risk factors for CRC, with 2.2 times higher risk of developing CRC compared with the general population [117], which is specifically called colitis-associated colorectal cancer (CAC), while ileal CD has to be surveilled with regard to SBA. An updated meta-analysis of population-based cohort studies has quantified the incidence of CRC among patients with IBD to be 1%, 2% and 5% after 10, 20 and >20 years of disease duration [118]. Another large meta-analysis assessing CRC risk in patients with IBD showed a risk of 2% at 10 years after UC diagnosis, 8% at 20 years and 18% at 30 years after colitis onset [119,120]. Taken together, CAC remains an important consequence of long-standing IBD, with an estimated incidence of approximately 5% after 20 years of disease duration [121]. Important clinical differences exist between CAC and sporadic CRC in the general population. The first is more common among young patients both in cases of UC and CD (average age of 50–60 years in IBD compared with 65–75 years for sporadic CRC in the general population) [122]; CAC is more likely to be found in the proximal colon (51.5%) compared to sporadic CRC (36.4%), especially in presence of primary sclerosing cholangitis (PSC) [123]. Furthermore, CACs are more commonly synchronous (15–20% of CAC compared with 3–5% of sporadic CRC), have an increased frequency of mucinous or signet ring cell histology and bear generally different genetic alterations [119,124,125]. The evolution of the epidemiology of CACs over the years seems to show a reduction in the incidence rate. This result might be attributed to the improvement of therapies for patients with IBD and to the advent of surveillance colonoscopy programs with early colectomy [122,126].

The principal risk factors for the development of CAC are: IBD diagnosis at young age (<15 years) and longer duration of the disease; male sex; extensive colitis; persistence and severity of the inflammation; and coexistence of PSC [121,127]. An important marker of disease severity and persistence of inflammation may be the development of colonic strictures. Recent studies suggest that 2% to 3.5% of colonic strictures harbor dysplasia or CRC [121,128,129]. Unlike sporadic CRC, usually occurring as the end point of the adenoma–carcinoma sequence, CACs follow the sequence inflammation–dysplasia carcinoma [122].

Chronic inflammation and the degree of immunosuppression are the main driving factors for IBD-related carcinogenesis, which is a process of clonal evolution [119]. IBD-associated inflammation has the potential to mediate clonal evolution over time, by mechanisms of induced by oxidative stress, inflammatory chemokines and cytokine (IL-6, STAT3, TNF-α, IL-10, IL-12 and IL-23) hyperproduction that affect numerous metabolic processes involved in cell repair, eventually creating a microenvironment that provides a selective advantage to those clones able to more rapidly repopulate the healing mucosa and to survive a cytotoxic inflammatory insult [119,130].

A proper understanding of genetic mutations should allow a better stratification of IBD patients according to their risk for dysplasia and invasive carcinoma, in order to personalize their treatment and surveillance; for example, a recent study found that architectural distortion seems to be significantly correlated with p53 and p21 overexpression in epithelial cells. Several studies have identified the tissue expression of specific proteins such as p53 and p21 in patients with IBD, in order to identify the natural evolution of these biomarkers and their relationship with carcinogenesis [119,130]. CACs have increased mutation frequencies of various other intracellular and intercellular signaling molecules, such as IL-16, which is overexpressed in IBD in an inflammation-dependent manner, or RADIL, a gene encoding a modulator of Rho GTPase signaling in cell migration, which might provide a selective advantage in mucosal healing [119]. Emerging studies in the field of microbiome analysis are revealing the role of the gut microbiota and intestinal barrier function in tumorigenesis, and animal studies are beginning to shed some light on the complex and dynamic interplay between the altered immune system, the aberrant gut microbiome and cancer development in IBD. Specifically, it was hypothesized that dysbiosis, and changes in population of microbial species including Fusobacterium nucleatum (*Fn*), Bacteroides or Prevotella, might enhance CRC progression by simultaneously regulating multiple signaling cascades that could lead to upregulation of proinflammatory responses, oncogenes, modulation of host immune defense mechanisms and suppression of DNA repair systems [131,132].

## 8. Micronutrients and Molecular Tuning of Colorectal Carcinogenesis

### 8.1. Vitamin D

Among the multiple factors involved in cancer development and progression, vitamin D is assuming an increasingly important role due to its pleiotropic effects [133].

Vitamin D comprises a group of fat-soluble secosteroids responsible for increasing intestinal absorption of calcium, magnesium and phosphate, and many other biological effects. In humans, the most important compounds in this group are vitamin D3 (also known as cholecalciferol) and vitamin D2 (ergocalciferol). Vitamin D’s influence on multiple biologic functions is expressed through the action of calcitriol, the product of a double hydroxylation of cholecalciferol, and the vitamin D3 receptor; therefore, aberrations in the physiological activity of Vitamin D may be a consequence of both its impaired serum concentration and defective receptor activity due to genetic mutations/variants [134].

The relationship between vitamin D and CRC has been explained both by epidemiologic studies evidencing low concentrations of the vitamin in subjects affected by cancer and by an alteration of its metabolic pathway in CRC tissues, although these findings do not have a clear clinical application yet [135]. Several studies have demonstrated its ability to interfere with cellular differentiation and proliferation both in normal and malignant tissues, with particular antiproliferative, proapoptotic, antimigration, anti-invasion, antiangiogenic and immunosuppressive activity in neoplastic cells [133,136]. The antiproliferative mechanism of vitamin D is due to the influence of calcitriol on cell cycle arrest in the resting phase G0/G1 by inducing the expression of the inhibitors of cyclin-dependent kinase, including p21, p27 and cystatin D, and stimulation of apoptosis [137,138,139].

Calcitriol was shown to upregulate miR-627, a ligand of the jumonji domain of histone demethylase, thus inhibiting the proliferation of CRC cells through epigenetic regulation in vitro and in vivo [139].

Vitamin D3 also promotes cell differentiation by increasing the expression of E-cadherin, cell adhesion proteins, alkaline phosphatase and maltase. Calcitriol is proved to inhibit β-catenin transcriptional activity in CRC cells, hence countering the aberrant activation of WNT-β-catenin pathway, which is the most commonly alternated signal pathway in sporadic CRC [140].

Moreover, the vitamin D receptor (VDR) inhibits cell proliferation and induces cell differentiation by binding to pi3k. Clinical trials showed that in *KRAS*-mutated/*PI3K*-mutated CRC tumor tissues, VDR was independently overexpressed [141]. Mocellin discussed epidemiologic data, suggesting a connection between vitamin D3 and cancer, and the results of clinical trials, which are conflicted [142]. Gandini et al. found that there was an inverse relationship between these levels and CRC [134,143].

The inhibition of angiogenesis was suggested in a paper by Pendas-Franco et al. that showed the ability of vitamin D to downregulate DKK-4, an antagonist of Wnt in CRC cells [144]; the same concept was also confirmed in papers by Meeker et al. and Shintani et al., who suggested vitamin D as anticancer agent due to its ability to inhibit growth of oral squamous cell carcinoma [145,146,147]. Antineoplastic roles of biologically active vitamin D3 includes the suppression of chronic inflammation, which indirectly inhibits cancer angiogenesis and invasion, and modulates the activity of factors related to cancer promotion (e.g., cyclooxygenase 2 (COX-2) and NF-kB). Another indirect evidence of anticancer properties of vitamin D is its role in the modulation of the immune response, and in particular inflammation [145,148]. Calcitriol may exert anti-inflammatory properties by inhibiting NF-kB signaling, the activation of which results in the production of proinflammatory cytokines [149,150]. Moreover, it may suppress p38 stress kinase signaling, therefore inhibiting the production of proinflammatory cytokines including IL-6, IL-8 and TNFα. Multiple studies have demonstrated the impact of vitamin D on lymphocytes CD4+ and CD8+, decreasing their proliferation, as well as on macrophages and dendritic cells, decreasing the secretion of proinflammatory cytokines after activation [145].

Although studies are limited, vitamin D has demonstrated to improve the cytotoxic activity of NK cells and the migration of dendritic cells into lymph nodes [151], overall modulating the immune response. The effects of active vitamin D are conveyed by its intracellular nuclear receptor VDR, the alterations and polymorphisms of which are responsible for an impaired activity of vitamin D. The *VDR* coding gene, located on the long arm of chromosome 12 (12q13-14), is associated with several SNPs, the most frequently studied being *FokI*, *BsmI*, *Tru9I*, *ApaI* and *TaqI*. Among them, the variation in *FokI* genotypes produces a smaller protein with increased activity. Several studies have demonstrated the association of the *VDR* polymorphisms with various diseases, including CRC [152,153], although results are still controversial and vary based on the considered population. A case–control study by Zhang et al. conducted in a Thai population failed to demonstrate significant associations between *VDR* SNPs and CRC, although a specific haplotype, AGGT, significantly predicted a lower risk of CRC [154]; moreover, the study found an interaction between dietary vitamin D intake and *VDR ApaI* genetic polymorphism in relation to the risk of CRC. A meta-analysis by Yu et al. suggested a moderate protective effect against CRC of the *VDR BsmI* polymorphism [155]. A study by Slattery et al. reported that the *FokI* (rs10735810), *BsmI* (rs11568820) and *CDX2* (rs11568820) polymorphisms of *VDR* were associated with *KRAS* mutation in CRC [156]. Clinical consequences of such a broad spectrum of regulations of cell cycle and differentiation have been evaluated in several epidemiological studies that aimed to clarify whether vitamin D deficiency can be considered a risk factor for CRC, or conversely if vitamin D physiological serum concentration and eventual supplementation may represent protective factors against CRC.

Over the last 20–30 years, several trials have been conducted, mostly finding a link between vitamin D deficiency and increased CRC risk and mortality [157,158,159], although other works could not confirm a statistical significance for this association. A meta-analysis by Lee et al. suggested an inverse association between circulating 25(OH)vitamin D levels and CRC (OR 0.77), with a stronger association for rectal cancer (OR 0.20) [160]. Similarly, a systematic review and meta-analysis by Yin et al. supported an inverse association between serum 25(OH)vitamin D and the risk of colon and rectal cancer, with odds ratios of 0.78 and 0.41, respectively [161]. Besides the potential role of vitamin D as a protective factor for CRC, other studies focused on its effects on the outcome of affected patients. A meta-analysis by Li et al., although including heterogeneous studies, confirmed that patients with the highest quartile of circulating 25(OH)vitamin D had a better overall survival compared to those in the lowest quartile [162]. With the aim to apply vitamin D as a prognostic marker for CRC patients, a recent study by Yuan et al. also investigated the relationships between plasma vitamin D binding protein (VDBP), bioavailable or free 25(OH)vitamin D and CRC survival, concluding that prediagnostic circulating concentrations of VDBP were positively associated with survival, while neither bioavailable nor free 25(OH)vitamin D levels were associated with overall or CRC-specific mortality [163]. Starting from these premises, other studies focused on the potential usefulness of vitamin D supplementation to improve CRC patient management. A systematic review with a meta-analysis of randomized controlled trials by Vaughan-Shaw et al. examined the effect of vitamin D supplementation on survival outcomes in patients with CRC, concluding that supplementation imparts a 30% reduction in adverse survival outcomes overall, with a 24% reduction in CRC-specific death and a 33% reduction in disease progression or death [164]. Overall, vitamin D seems to have a promising role as a prognostic factor for CRC patients’ outcome and an easy element to improve in case of deficiency, being widely available and cheap to apply in large populations at all ages.

### 8.2. Calcium

Strictly related to vitamin D, calcium has also been explored as a molecule impacting on CRC risk. Being a ubiquitous second messenger, and signaling for a variety of cellular processes such as control of the cell cycle, apoptosis and migration, calcium activates a variety of ion-specific channels, cotransporters and pumps. The expression of several genes coding for calcium channels has demonstrated to be upregulated in CRC cells, including *TRPC1* and *TRPM2* [165,166], the activity of which has been related to the promotion of metastases; while *TPRM6*, the expression of which has been related to better patient survival, has been found to be downregulated in CRC cells [167]. Moreover, stromal interaction proteins 1 and 2 were revealed to be up- and downregulated in CRC, respectively, causing increased CRC cell motility and apoptosis resistance [168,169]. Besides regulating cell signaling, clinical applications of calcium supplementation with diet and cancer risk or progression have also been explored. Although cancer proliferation has been associated with an upregulation of calcium [170], Garland et al. found that a calcium-rich diet reduced the risk of CRC [171]. A systematic review of randomized controlled trials found that calcium supplementation with doses from 1200 to 2000 mg/day and treatment duration from 36 to 60 months reduced the risk of recurrent colorectal adenomas (RR = 0.89, 95% CI: 0.82–0.96, 5 studies, 2984 participants) [172]. It was proposed that calcium binds bile acids in the bowel lumen, inhibiting their proliferative and carcinogenic effects [173]. In support of this hypothesis, studies in animals have indicated a protective effect of dietary calcium on bile-induced mucosal damage and experimental bowel carcinogenesis [174].

Although the biochemical and the clinical behavior of calcium with regard to CRC seem contrasting, calcium signaling promotes or inhibits cancer based on the ability of the tissue environment to maintain balance of its intra- and extracellular concentrations: the increase of intracellular calcium promotes cancer progression, but once the level has reached overload, cancer cell death is favored, deteriorating cancerous tissue. Although the clinical application of such behavior is not yet available, calcium channels may present as possible drug targets to reduce tumor burden [175].

## 9. Attempting Pharmacological Interference with CRC Development: Chemoprevention

In the last decades, the growing knowledge about the physiopathology of CRC and its molecular players has allowed researchers to shed light on the potential application of drugs as preventive tools. Chemoprevention refers to the long-term use of a variety of oral medications that can delay, prevent or even reverse the development of colonic adenomas, and interfere with the multistep progression from adenoma to carcinoma.

Focusing on IBD patients, the use of maintenance therapies, and notably the better control of inflammation by improved medical therapy and higher rates of mucosal healing, could be important strategies for reducing CRC risk in UC patients [176]. Literature data about the preventive effect of specific drugs on the development of CAC are scarce; moreover, the available studies are focused on the use of the first molecules used for the treatment of IBD, while long-term trials about the effect of biologic therapies are awaited. 5-ASA is a first-line agent for IBD therapy. This molecule is able to reduce oxidative stress, inhibit cell proliferation and promote apoptosis. Most reports indicated that 5-ASA reduces the risk of CRC in UC, although literature data are controversial [177,178]. This protective effect has also been studied in CD; a study by Cahil et al. concluded that the use of salicylates is protective against SBA [179]. Overall, the protective effect of immunomodulators is primarily due to their role in the control of inflammation [180]. Ursodeoxycholic acid (UDCA) may be a practical chemoprevention against colonic exposure to bile acid in patients with PSC. UDCA reduces the colonic concentration of the secondary bile acid as a carcinogen [126]. Given the known importance of TNF and interleukins within the pathogenesis of CAC, more targeted inhibition of these pathways may offer an opportunity to prevent CAC, particularly among high-risk individuals who have developed early dysplastic lesions, where these cytokines serve to stabilize the cancer microenvironment. Animal models have suggested that TNF antagonists may prevent the development or progression of dysplasia and cancer, and some population-based data within IBD have demonstrated a lower frequency of CRC among those treated with infliximab.

Although the role of anti-inflammatory agents as chemopreventive drugs is crucial in CAC, these medications have been considered for sporadic and hereditary CRC for decades [181].

Aspirin has been the first extensively investigated drug in the chemoprevention of colorectal adenomas and cancer, thanks to its ability to inhibit COX-1 and COX-2 enzymes, both of which are important mediators of prostaglandin production. In 1988, a population-based case–control study by Kune et al. demonstrated that regular aspirin users showed a relative risk of 0.53 of developing CRC, compared with nonconsumers [182]. Since then, several large studies have been developed, agreeing on the protective role of aspirin against CRC [183,184]. Unfortunately, aspirin has several well-known side effects, including gastrointestinal hemorrhage, renal toxicity, and risk of developing Reye’s syndrome [185] or Stevens–Johnson syndrome. Being that most side effects of aspirin and NSAIDs in general are related to their inhibition of COX-1, selective drugs to inhibit COX-2 have been developed and applied not only in the treatment of inflammation, but also in chemoprevention of CRC, also justified by the demonstration of an overexpression of COX2 in adenomatous lesions [186]. In particular, hereditary syndromes at risk of developing CRC have been addressed, including FAP and Lynch syndrome [187]. In 2000, a double-blind, placebo-controlled study by Steinbach et al. conducted on 77 patients affected by FAP demonstrated a significant reduction of the number of polyps after 6 months of treatment with oral celecoxib [187]. A recent double-blind, placebo-controlled trial by Burn et al. reporting a 10-year follow-up of 861 patients affected by Lynch syndrome demonstrated a significantly reduced risk of developing CRC for aspirin consumers compared to the placebo group (HR 0.65), with a similar adverse-events rate between groups [188]. Overall, evidence is in favor of NSAIDs’ long-term use in the case of hereditary syndromes at risk of developing CRC, although this type of chemoprevention is not yet uniform nor systematically used worldwide, probably because the risk/benefit ratio and the optimal dosing have not yet been standardized.

## 10. Microbial Hosts: Fusobacterium Nucleatum

Considering the environmental factors potentially participating in CRC onset and progression, in the last decades, growing attention has been paid to the role of the intestinal microbiota alterations. Among bacteria, *Fn* may contribute to CRC development through multiple mechanisms, including the interaction with the host immune system, the production of cancer-associated metabolites and the release of genotoxic virulence factors [189,190]. The protumorigenic role of *Fn* and its association with CRC are supported by several studies and experimental models [191,192].

First, *Fn* has demonstrated to be enriched in CRC lesions compared to matched normal colonic mucosa; moreover, *Fn* sequences were found in lymph node and distant metastases [193,194]. The cancerogenic mechanisms of *Fn* start from the adhesion and invasion of the enterocytes by the bacterium, thanks to adhesion molecules (FadA and Fap2) able to recognize epithelial cells. After adhesion, *Fn* activates the β-catenin and NF-kB signaling pathways [195] as FadA-cadherin-E binding accelerates carcinogenesis in the presence of genetic alterations by beta-catenin activation; moreover, Fap2-TIGIT binding promotes tumor survival by smoldering antitumor immunity [196]. However, *Fn* cannot be yet considered a carcinogen *per se*, but rather a promoter of cancer progression in cells already altered by an initiating factor [192,197]. Studies also suggested than *Fn* could trigger EMT in the neoplastic colonic cells, promoting proliferation and invasion by enhancing the expression of EMT-related genes (E-cadherin and N-cadherin) [198,199].

Besides its mechanisms of action, *Fn* seems to play a double-faceted role in CRC progression and clinical behavior. Although *Fn* enrichment in stool or epithelial samples is associated with mucosal degeneration, presence of metastases [193,194,200] and chemoresistance [201] and increased risk of disease-specific mortality [202,203], *Fn*-positive CRCs are more frequently characterized by microsatellite instability [202,204,205], a group of tumors classified as usually having better prognosis than their counterpart microsatellite stable CRC, due to their higher immune infiltrate (TILs) and low metastatic potential [25,110,206]. Therefore, *Fn* could be intended as an accelerator of the carcinogenesis process and a modifier of cancer clinical behavior in a specific subset of tumors, namely MSI cancers. However, most recent data also implicate *Fn* in the responsiveness of locally advanced rectal cancers to neoadjuvant therapy. In such a setting, the persistence of *Fn* after therapy was associated with a worse outcome in two independent studies from Europe and Asia [207,208]. Accordingly, the role of *Fn* in CRC might be wider and more relevant than previously thought. Noticeably, large differences in study methodologies may account for discrepancies in findings so far, and the field requires appropriate validations in different clinical–pathological settings.

## 11. The Frame for Biomarker Actionability: TNM Staging System Turning in the 21st Century

The stage of cancer by the TNM system describes its advancement based on its local extent at the site of origin (T), coupled to the presence or absence of the involvement of the regional lymph nodes (N), and eventually of metastases at distant sites (M) [209]. The TNM continues to represent the cornerstone prognostic system for solid malignancies, although the American Joint Committee on Cancer (AJCC) has increasingly acknowledged the necessity to move toward individualized, more precise outcome estimates, mainly through the application of accurate risk models and calculators [210,211] incorporating nonanatomic prognostic features. Regarding CRC, in the latest AJCC 8th edition [209], published in 2016, particular emphasis has been given to MMR deficiency sustained by germline and somatic mutations or epigenetic changes, as well to RAS pathway mutations (i.e., *KRAS*, *BRAF* and *NRAS*). Still, a key drawback of risk calculators is the incapability to convey with heterogeneity within each stage groups.

The spread of cancer cells from the primary tumor to tumor-draining lymph nodes defines stage III CRC disease, and is the most relevant prognostic factor triggering the administration of adjuvant chemotherapy. The relationship between lymph nodes and distant metastases has been acknowledged since the 19th century, and together with the finding that lymph node disease frequently precedes systemic disease, has since then prompted the conception that surgical resection of positive lymph nodes may decrease the rate of recurrence. However, results from clinical trials have suggested that lymph node resection does not always increase patient survival [212], rousing the different notion that lymph node metastases do not necessarily imply distant metastatic spread [213]. This alternative view could be in line with the wide variability in survival rates within stage III CRC, ranging between 70% for T1N1a and 10–15% for T4bN2b tumors [214,215], despite adjuvant chemotherapy. A pooled analysis of more than 12,000 stage III CRC patients enrolled in the IDEA trial confirmed the large variability of five-year disease-free survival (DFS) within 16 substages based on T and N categories, ranging from 89% for T1N1a to 31% for T4N2b CRC [216]. Interestingly, the analysis also evaluated the contribution of each therapeutic option across the different substages. The authors used a metaregression model to estimate the five-year DFS within each T and N subgroup. While the projected five-year DFS for T1N1 cancer patients treated with surgery alone was 79.6%, patients with T4N2b disease showed a 13.9% five-year DFS with surgery alone, with an additional 11.2% absolute gain with adjuvant fluoropyrimidines alone, an additional 6.4% with oxaliplatin for three months and 2.5% with oxaliplatin for six months (Figure 2). These data underline the existence of distinct prognostic categories within stage III CRC contemporarily, implying a reappraisal of the bases of current treatment strategies. Likewise, a better interpretation of the link between lymph node involvement and the development of distant metastases is pivotal, considering the changes related to empirical treatment strategies.

The conceptual framework of the TNM staging system is seeded in the concept of the sequential progression of metastatic cascade, in which cancer cells from the primary tumor (T) seed local lymph node dissemination (N) that may eventually lead to metastases at distant sites (M) [217]. Metastatic spreading has been depicted in several genome sequencing studies that have revealed a clonal evolution of cells from primary tumor to metastatic sites [218,219]. Conversely, another view would posit that cancer cells can spread as early as from preneoplastic lesions [220,221] and from early-stage primary tumors, the vasculature abnormalities of which favor the escape of cancer cells into the circulation [222]. At any event, it remains unclear whether a distinct metastatic subclone develops in the primary tumor, afterwards disseminating to lymph nodes and distant sites [223,224,225], or whether multiple subclones in the primary tumor separately scatter lymphatic and distant metastases [219,226,227]. To evaluate the evolutionary origin of lymphatic and distant metastases, Naxerova and colleagues [228] studied 213 CRC specimens from 17 patients, showing that in up to 70% of the cases, lymphatic and distant metastases developed from independent subclones in the primary tumor. Thus, in the majority of patients, lymphatic and distant metastases might have an independent origin. Still, around 30% of cases shared a common subclonal origin.

## 12. Molecular Heterogeneity and Metastatic Seeding

Besides the timing of cancer cell spreading, it remains largely unclear how cancer cells develop the capability to colonize distant tissues. This ability may arise during primary tumor growth as a consequence of intrinsic properties of the tumor cells and of (faulty) host response, or as the effect of the selective pressure on previously spread cancer cells to adapt to distant tissue microenvironments [229]. Interaction among tumor cells themselves, as well as between host and tumor cells, can cause alterations in their behavior and plasticity. For example, hypoxia may exert a negative selection against *RAS*-mutant clones through a mechanism identified as secretory senescence [230]. In addition, *KRAS*-mutant senescent cells can then induce the development of RAS wild-type subpopulations by a paracrine mechanism, leading to their progressive outgrowth [231,232].

A better understanding of the biology of the development of metastases and of the properties of the cells selected along this process is critical for precision medicine and treatment selection for patients with systemic disease. Still, identification of the hallmarks of metastatic potential has been complex due to heterogeneity among tumor cells [233]. Throughout primary tumor evolution, abnormal levels of genetic instability lead to the development of cells with newly acquired features [233,234]. Several studies have assessed the genetic and phenotypic diversity of the tumor cells that encompass primary tumors [235]; nevertheless, the level of genetic and epigenetic heterogeneity and phenotypic plasticity below metastatic growth remains undefined. Single-cell and sequencing data indicate that some metastases develop from separate lineages [228,236,237,238,239,240], and metastases themselves can generate other metastases [236,241]. Hence, heterogeneity is part of an evolutionary and temporal process [242], yet it has a critical role in drug resistance and disease progression by preventing efficacy of single targeted therapy (Figure 3). As concerns CRC, Ciardiello and colleagues [243] have depicted specific molecular alterations differing among cancers (i.e., intertumor heterogeneity), as well as the presence of cancer cells with distinct molecular alterations within the same tumor sample (i.e., intratumor heterogeneity).

## 13. Molecular Heterogeneity and the Emergence of Resistance to Target Treatment in Metastatic CRC

Mutations along the RAS pathway are responsible for both primary and acquired resistance to anti-epidermal growth factor receptor (EGFR) therapies [230,244,245,246,247,248,249,250,251]. In various cases, *RAS* mutations arise early during CRC carcinogenesis, as a clonal (truncal) mutation maintained in primary and metastatic lesions [9,252], and *RAS*-mutant tumors are unresponsive to anti-EGFR therapies. Still, notwithstanding stringent selection based on screening for somatic *RAS* mutations, about 65–70% of patients progress within three to 12 months after initial anti-EGFR therapies. Analysis of post-treatment samples has revealed acquired resistance as a major limitation of therapies targeting oncoproteins such as EGFR and BRAF [253]. Seminal studies on plasma-cell-free DNA have shown that under drug selective pressure, undetectable *RAS*-mutant subpopulations at baseline undertake a clonal expansion, preceding acquired therapy resistance [254,255,256]. The clinical managing of patients who acquire *RAS* mutations subsequent to EGFR inhibition is doubtful. At progression, the majority of patients receive further lines of therapies based on chemotherapy alone or combined with antiangiogenic drugs, and eventually a monotherapy with the multikinase inhibitor regorafenib. Siravegna and colleagues [256] showed that *KRAS*-mutant alleles, which develop at the time of disease progression, decline when anti-EGFR treatment is interrupted, persisting under the limit of detection across succeeding lines of treatment. The decline of *KRAS*-mutant alleles detected in blood from patients after interruption of the anti-EGFR blockade [257] suggests not only a dynamic evolution of cancer cells, but also that a rechallenge therapy may be a clinically valuable choice in these patients, as CRC secondary lesions are likely to respond to anti-EGFR rechallenge [258].

Other changes can occur under the pressure of treatments. Drug-tolerant cancer cells that survive EGFR/BRAF inhibitor treatment show a decreased expression of mismatch and homologous recombination (HR) proteins, and increase their mutagenic rate [259]. All these alterations may trigger the RAS–MEK–ERK pathway [246,260,261,262]. Therefore, though resistance to anti-EGFR inhibitors can be polyclonal, it mostly converges on the downstream signaling pathways of EGFR [253]. In addition, the efficacy of monoclonal antibodies targeting a single pathway has been mainly limited by the occurrence of compensatory feedback loops in other pathways, such as increased secretion of vascular endothelial factor (VEGF) during anti-EGFR treatment [263].

The molecular heterogeneity detectable following anti-EGFR therapy emphasizes how a single therapeutic approach is unlikely to overwhelm extensive mechanisms of resistance, as most of these alterations involve multiple pathways in a single patient. Hence, the picture of tumor heterogeneity at the time of secondary resistance, as depicted for EGFR inhibitors, indicate that multitargeted drug combinations before relapse could better target the bulk tumor cells and reduce the expected acquired resistance mechanisms, thus providing a substantial improvement in survival compared with administration at progression [264,265].

## 14. Restraining the Progression of Metastatic CRC: The Frontier

The latest scientific enhancements of molecular diagnostics; i.e., blood-based tumor genotyping, have permitted the assessment of clonal evolution in patients with cancer, and introduced the new concept of time, to guide adaptive therapy strategies.

Regorafenib is an oral multikinase inhibitor approved by both the Food and Drug Administration and the European Medicines Agency for CRC patients who have not responded to available therapies [266]. It inhibits three oncogenic pathways, specifically: (a) cell growth by inhibition of KIT, RET, RAF-1 and BRAF; (b) tumor angiogenesis by targeting vascular endothelial growth factor receptors (VEGFR) 1, 2 and 3, and the tyrosine kinase with immunoglobulin and EGF homology domain 2 (TIE2); and (c) the tumor microenvironment by hampering fibroblast growth factor receptor (FGFR) and platelet-derived growth factor receptor-b (PDGR-b) [267,268,269]. The combined treatment with cetuximab and regorafenib prompts synergistic antiproliferative and proapoptotic effects by blocking MAPK and AKT pathways both in vitro and in vivo [270], and is a potential approach worth exploring in an attempt to overwhelm primary or secondary resistance to EGFR inhibitors in patients with advanced CRC. The results of the REVERCE randomized phase II trial suggest that the sequence of second-line regorafenib followed by cetuximab/irinotecan in CRC after failure of fluoropyrimidine, oxaliplatin and irinotecan is associated with a longer survival compared with the standard sequence of cetuximab/irinotecan followed by regorafenib [271]. Biomarker analyses have revealed earlier occurrence of changes in *RAS, BRAF, EGFR, HER2* and *MET*, commonly associated with resistance to anti-EGFR therapy [246,255,272,273] after cetuximab compared with regorafenib, thus explaining the poorer outcomes with cetuximab in the first treatment arm compared with regorafenib given first. The randomized REVERCE II trial (NCT04117945) comparing regorafenib followed by anti-EGFR monoclonal antibody therapy versus the reverse sequencing for metastatic CRC patients formerly treated with fluoropyrimidine, oxaliplatin and irinotecan is currently ongoing, and will probably provide further data concerning the optimal sequence of treatments.

The expected utility of liquid biopsy in this setting is to identify the circulating clonal background in cancer patients through the analysis of circulating tumor DNA, providing innovative and clinically meaningful understandings of tumor heterogeneity sustaining drug resistance [274]. Acquired resistance to EGFR-targeted monoclonal antibodies has been extensively associated with the emergence of RAS pathway mutations detectable in the blood of patients before the appearance of clinically manifest disease progression [254,257,275]. Contrariwise, the selective pressure exerted by antiangiogenic drugs in CRC patients with *RAS*-mutant disease has been less frequently examined. Liquid biopsy under antiangiogenic treatment has revealed the relative prevalence of *RAS* wild-type clones, which can be translated in a clinically significant advantage for patients. Targeting this gap with EGFR inhibitors potentially could provide an available second-line choice in *RAS*-mutant CRC. The KAIROS trial (Keeping the Advantage of the Impermanent RAS–Wild Type Window Offering Second-Line EGFR Inhibitors, EudraCT Number 2019-001328-36) may help to establish whether the response to EGFR blockade, in patients with *RAS*-mutant primary tumors could switch to *RAS* wild-type clones during first-line antiangiogenic therapy.

Over the decades, the vision on CRC has tremendously changed. The application of genetics, NGS, advances in immunology and the understanding of the value of TME, micronutrients and the microbiome are leading to a deeper understanding of the multifaceted behavior and subtypes of CRC, providing the bases for precision medicine, with the aim to improve the patient’s outcome.

## Figures and Tables

**Figure 1 ijms-22-05246-f001:**
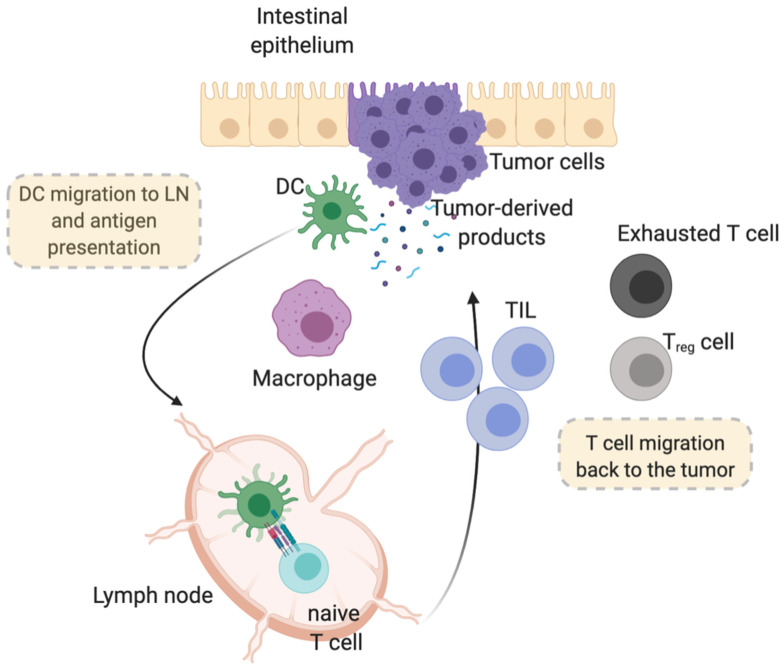
Scheme of the cancer immune cycle, depicting antigen-presenting cells (mostly dendritic cells, DC), which infiltrate the tumor tissue, uptake tumor-derived products and traffic to draining lymph nodes, therein presenting antigens to antigen-specific cytotoxic T cells. Sustained immunosuppressive circuits may induce T-cell dysfunction, immune escape and eventually cancer progression.

**Figure 2 ijms-22-05246-f002:**
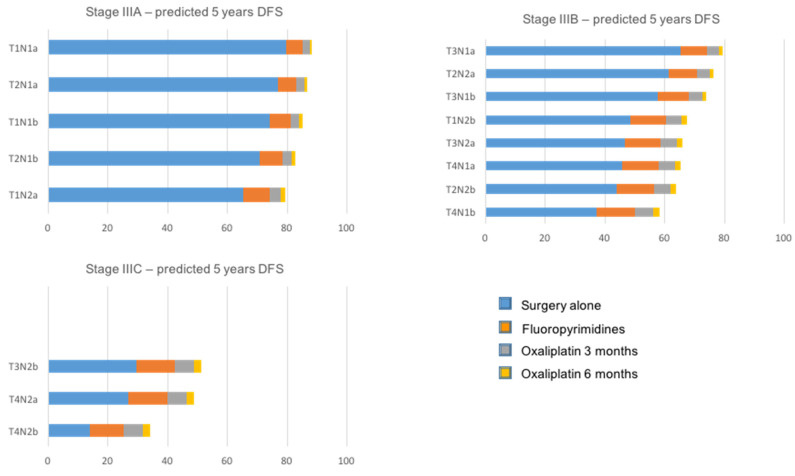
Prognostic subgroups within stage III colon cancers by therapeutic options: surgery alone; fluoropyrimidine alone; oxaliplatin-based doublet for 3 months; oxaliplatin-based doublet for 6 months.

**Figure 3 ijms-22-05246-f003:**
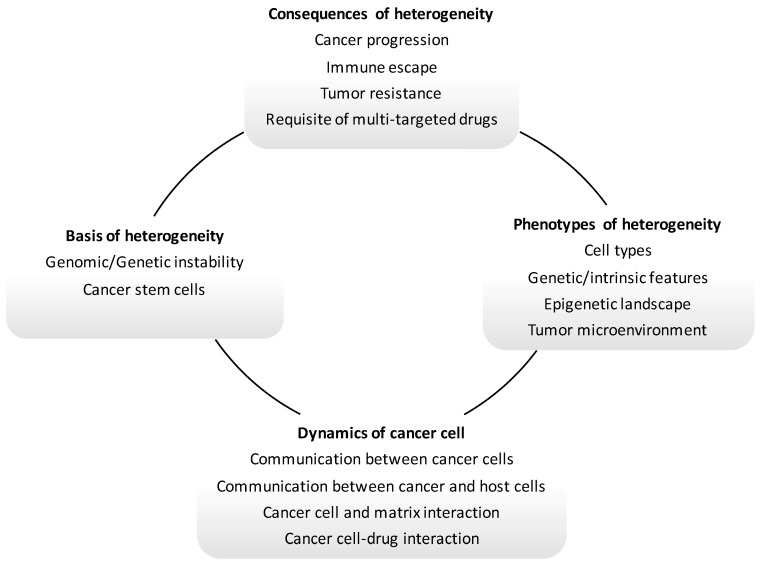
Principles of temporal and spatial heterogeneity of cancer.

**Table 1 ijms-22-05246-t001:** Molecular phenotypes of colorectal cancer according to the prevalent patterns of alterations.

DNA		mRNA	
Type of Gene Damage	Methylation	Gene Expression Patterns	
Microsatellite instability (MSI) = mismatch repair (MMR) deficient	CpG island methylator (CIMP+)	Consensus molecular subtype (CMS) 1 CRC intrinsic subtype (CRIS)-A/B	
Microsatellite stable (MSS) = MMR proficient = chromosomal instability (CIN)	Mostly CIMP-	CMS2, canonical	CRIS-C
		CMS3, metabolic	CRIS-D
		CMS4, mesenchymal	CRIS-E/B

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
