# Peer review of "Heterogeneity of Colorectal Cancer Progression: Molecular Gas and Brakes"

_ijms, 2021, doi:10.3390/ijms22105246_

Round 1

Reviewer 1 Report

The manuscript reviews the present progress in colorectal genetics, treatment and survival together  with the broad aspects of other bowel diseases which can increase the risk of CRC. The colorectal cancer is a very important issue since this is the fourth most frequent malignancy diagnosed in humans. It is also difficult for the effective treatment, especially for more advanced cases. The authors described a long list of topics connected with colorectal cancer including different aspects of CRC genetics, immune response, inflammation, tumor microenvironment, attempts of chemoprevention, microbiome and molecular basis of the resistance to treatment.

The paper is interesting so the reader can get the basic information about the plethora of aspects in colorectal carcinogenesis together with the literature in the particular field.

Author Response

The authors are thankful to the reviewer for the evaluation of the paper and for having appreciated our effort to describe such a broad and actual topic.

---

Reviewer 

The manuscript reviews the present progress in colorectal genetics, treatment and survival together  with the broad aspects of other bowel diseases which can increase the risk of CRC. The colorectal cancer is a very important issue since this is the fourth most frequent malignancy diagnosed in humans. It is also difficult for the effective treatment, especially for more advanced cases. The authors described a long list of topics connected with colorectal cancer including different aspects of CRC genetics, immune response, inflammation, tumor microenvironment, attempts of chemoprevention, microbiome and molecular basis of the resistance to treatment.

The paper is interesting so the reader can get the basic information about the plethora of aspects in colorectal carcinogenesis together with the literature in the particular field.

Answer: The authors are thankful to the reviewer for the evaluation of the paper and for having appreciated our effort to describe such a broad and actual topic.

Reviewer 2 Report

In this review, Gaiani et al authors summarized current understanding on heterogeneity of colorectal cancer (CRC), including the application of genetics, NGS, advances in immunology, the value of TME, micronutrients and microbiome in CRC progression. They also briefly described current progress on the treatment of colorectal cancer with chemoprevention and targeted therapy. Overall, the manuscript is well written and provides a comprehensive knowledge on CRC. One concern is that the review is not easy to follow since there is no connection among sections. There are also some typos have to be corrected, such as line 221 “NF-Kb”

Author Response

The authors thank the reviewer for the evaluation of the paper and for his suggestions. The paper has been revised and sentences of connection through paragraphs have been added to clarify the line of discussion. Moreover, all typos have been corrected, including the one suggested by the reviewer.

---

Reviewer

In this review, Gaiani et al authors summarized current understanding on heterogeneity of colorectal cancer (CRC), including the application of genetics, NGS, advances in immunology, the value of TME, micronutrients and microbiome in CRC progression. They also briefly described current progress on the treatment of colorectal cancer with chemoprevention and targeted therapy. Overall, the manuscript is well written and provides a comprehensive knowledge on CRC. One concern is that the review is not easy to follow since there is no connection among sections. There are also some typos have to be corrected, such as line 221 “NF-Kb”

Answer: The authors thank the reviewer for the evaluation of the paper and for his suggestions. The paper has been revised and sentences of connection through paragraphs have been added to clarify the line of discussion. Moreover, all typos have been corrected, including the one suggested by the reviewer.
